# Plasmonic Sensors beyond the Phase Matching Condition: A Simplified Approach

**DOI:** 10.3390/s22249994

**Published:** 2022-12-19

**Authors:** Alessandro Tuniz, Alex Y. Song, Giuseppe Della Valle, C. Martijn de Sterke

**Affiliations:** 1Institute of Photonics and Optical Science (IPOS), School of Physics, The University of Sydney, Sydney, NSW 2006, Australia; 2University of Sydney Nano Institute, The University of Sydney, Sydney, NSW 2006, Australia; 3School of Electrical and Information Engineering, The University of Sydney, Sydney, NSW 2006, Australia; 4Dipartimento di Fisica, Politecnico di Milano, Piazza Leonardo da Vinci 32, 20133 Milan, Italy; 5Istituto di Fotonica e Nanotecnologie, Consiglio Nazionale delle Ricerche, Piazza Leonardo da Vinci 32, 20133 Milan, Italy

**Keywords:** plasmonics, sensors, fibre sensors, coupled mode theory, hybrid plasmonic waveguides, directional coupling, photonic integrated circuits

## Abstract

The conventional approach to optimising plasmonic sensors is typically based entirely on ensuring phase matching between the excitation wave and the surface plasmon supported by the metallic structure. However, this leads to suboptimal performance, even in the simplest sensor configuration based on the Otto geometry. We present a simplified coupled mode theory approach for evaluating and optimizing the sensing properties of plasmonic waveguide refractive index sensors. It only requires the calculation of propagation constants, without the need for calculating mode overlap integrals. We apply our method by evaluating the wavelength-, device length- and refractive index-dependent transmission spectra for an example silicon-on-insulator-based sensor of finite length. This reveals all salient spectral features which are consistent with full-field finite element calculations. This work provides a rapid and convenient framework for designing dielectric-plasmonic sensor prototypes—its applicability to the case of fibre plasmonic sensors is also discussed.

## 1. Introduction

Waveguide sensors which use surface plasmon polariton (SPP) resonances [1] are particularly attractive for bio-sensing at the nanoscale [2,3,4,5,6,7]. Such sensors harness the deep subwavelength lateral confinement of SPPs to characterise small modifications to a nanoscale environment via changes in the propagating field’s phase or loss. Originally implemented using free-space bulk optics (e.g., in the Kretschmann [8] and Otto [9] configurations), SPP sensors are ideal for integration with chip-scale [10,11,12,13,14] and fibre-based [15,16,17,18] platforms, providing a monolothic and convenient way of detecting small changes near the metal surface—see for example Refs. [19,20,21] as a selection of recent reviews.

When designing any refractive index sensor, one of the most important aspects to consider is how resonant spectra—characterized by a transmission minimum at a wavelength $\lambda_{R}$—change with the refractive index $n_{a}$ of an analyte. A sensor’s overall performance is often defined in terms of its detection limit (DL), i.e., the smallest detectable change in refractive index δn, and which generally depends on a specific user’s experimental configuration. It can be shown that [22,23]
(1)$$\delta n\propto \frac{\delta \lambda }{S},$$
where δλ is a characteristic spectral width (typically taken as the Full Width at Half Maximum (FWHM) [24,25]), and the sensitivity $S=d\lambda_{R}/dn_{a}$ quantifies the shift in the resonance minimum with analyte index. The smallest detectable δn thus stems from a combination of narrow spectral width and high sensitivity. The commonly used figure of merit (FOM) [20], which should be maximized during sensor optimization, is given by the inverse of the right hand side of Equation (Equation 1):(2)$$\mathrm{FOM}=\frac{S}{\delta \lambda }.$$

The transmission spectra used to maximize Equation (Equation 2) typically have minima at the phase matching (PM) condition between dielectric and plasmonic modes [26]. The shift of the PM condition as $n_{a}$ varies, thus often provides a first estimate of the sensitivity. However, we wish to emphasize that, in practice, the phase-matching condition in isolation provides insufficient information to infer the spectral minima, and can even lead to incorrect conclusions. In fact, the physics behind such resonant spectra is quite complex [25], due to the subtle and often counter-intuitive interplay of phase matching, modal coupling, interference, and losses. To highlight this subtlety, it is useful to revisit the textbook example [27] of possibly the simplest plasmonic sensor implementation, based on the Otto plasmonic coupler [9], shown schematically in Figure 1a. In this particular scheme, a plane wave at wavelength λ= 800 nm is incident from the top of a semi-infinite silica prism (refractive index: $n_{s}=1.5$) towards its bottom surface, at an angle θ with respect to the normal. The spacer (green) is the analyte. In the absence of a metal, and for $n_{a}<n_{s}$, total internal reflection leads to an evanescent field at the silica/analyte interface. By introducing a gold layer (refractive index: $n_{Au}=0.23+4.5i$), spaced a distance *w* from the edge of the prism, the evanescent field can excite a bulk SPP if momentum is conserved, which occurs at an angle $\theta_{\mathrm{SPP}}$. In the first instance, the SPP excitation angle can be calculated analytically by recalling that the bulk SPP has a propagation constant of
(3)$$\beta_{\mathrm{SPP}}=\frac{2\pi }{\lambda }\sqrt{\frac{\varepsilon_{a}\varepsilon_{\mathrm{Au}}}{\varepsilon_{a}+\varepsilon_{\mathrm{Au}}}},$$
and that the wavevector component parallel to the surface of the prism is given by
(4)$$\beta_{\left|\right|}=\frac{2\pi }{\lambda }n_{s}\sin\theta_{\mathrm{SPP}},$$
so that momentum conservation (i.e., $ℜe\left(\beta_{\mathrm{SPP}}\right)=\beta_{\left|\right|}$) leads to
(5)$$\theta_{\mathrm{SPP}}\approx \sin^{-1}\left(\frac{1}{n_{s}}\sqrt{\frac{\varepsilon_{a}\varepsilon_{\mathrm{Au}}^{\prime }}{\varepsilon_{a}+\varepsilon_{\mathrm{Au}}^{\prime }}}\right),$$
where $\varepsilon_{\mathrm{Au}}^{\prime }=ℜe\left(\varepsilon_{\mathrm{Au}}\right)$. In the case of air, $n_{a}=1$ leads to $\theta_{\mathrm{SPP}}\approx 43.1^{\circ }$; for water, $n_{a}=1.33$ and $\theta_{\mathrm{SPP}}\approx 68.2^{\circ }$. Note that these results do not depend on the spacer thickness *w*. However, the efficiency of energy transfer from the incident light to the SPP, and from the SPP back into the radiation field of the prism, also depends on the coupling strength between the two evanescent modes via separation *w*, and therefore phase matching only provides a partial description of sensor performance. To highlight this, we calculate the reflectivity *R* using the Fresnel equations [27] for different values of *w*, and show the results in Figure 1b for $n_{a}=1$. The narrowest reflectance spectrum, corresponding to the largest FOM for this $n_{a}$, and associated with the highest SPP coupling efficiency, occurs only for a specific w=1000nm, at $\theta_{\mathrm{SPP}}=43.0^{\circ }$—which itself differs from the Equation (Equation 5) prediction. This example serves to illustrate and highlight the importance of considering propagation through a specific device configuration when designing refractive index sensors. Figure 1b also shows that, at other values of the spacing *w*, the dip in transmission is shallow, leading to inferior sensor performance. A full calculation of the reflectance as a function of *w* and θ for $n_{a}=1$ and $n_{a}=1.33$ is shown in Figure 1c and Figure 1d respectively: the phase matching condition, shown as black dashed lines, leads to incorrect predictions of the spectral minima for some configurations, with the full spectrum being highly dependent on the choice of *w*. Thus, even in this simple, bulk geometry, knowledge of the phase matching angle $\theta_{\mathrm{SPP}}$ provides insufficient information for the design of a sensor with high sensitivity.

We now turn to plasmonic waveguide sensors, which are the focus of this work. In this case, sensor performance is based on directional coupling [28,29], which also relies on resonant energy transfer between waveguides, and is used for sensing applications in several different contexts [22,24,30,31]. Compared to the Otto configuration, hybrid plasmonic waveguide couplers provide a pathway for photonic circuit integration, as well as more localized confinement and higher spatial resolution. Figure 2 shows a schematic of an example of a chip-scale hybrid plasmonic waveguide coupler, which we will use as the example platform to illustrate our method. We consider a one-dimensional slab device supporting two-dimensional propagation, grounding our theoretical discussion to a realistic device which we can also use to compare with full numerical calculations. Such a waveguide coupler is described by many parameters, i.e., refractive indices, width, spacing, and wavelength. Here, the dielectric waveguide (purple) is taken to be a silicon slab (width: d=220nm, refractive index: 3.5); the metal waveguide (yellow) is taken to be gold (thickness: t=7.5nm; with the permittivity following from the Drude model [32]; edge-to-edge separation: s=400nm). The background is silica [33] and the region above the gold is covered by an analyte (refractive index range: $n_{a}=1.3-1.5$). Light is injected into the dielectric core, which in turn couples to the two modes in the sensor region, yielding an overall transmission spectrum that depends on including the wavelength λ, length *L*, and refractive index $n_{a}$.

Analogously to the Otto configuration, modal calculations alone (e.g., which monitors the numerically calculated phase-matching wavelength, or loss-matching wavelength) are insufficient for predicting how a sensor will perform [25,34,35]. All approaches used so far to achieve this rely on knowledge of the electric and magnetic fields, and calculating mode overlap integrals [25,29,36,37,38,39], which can be cumbersome when designing multi-material two-dimensional waveguides with fine feature sizes, as is often the case in plasmonic sensors. Full field propagation methods, such as Finite Difference Time Domain and Finite Element methods, are much more computationally demanding, particularly in three dimensions, where length scales associated with cross sections and propagation distances can differ by many orders of magnitude.

Spatial coupled mode theory (CMT) approaches [40], in contrast, are far simpler: they require knowledge of just a small set of reduced parameters which account for propagation and coupling between the waveguide modes. While CMT can lack quantitative accuracy [41], it provides rapid and immediate intuition of coupled waveguide performance, with typically excellent agreement with full simulations [40]. Here, we present a simplified CMT approach for lossy directional couplers, via an easy-to-implement perturbation of the lossless case, which can be used to predict the performance of a full hybrid plasmonic waveguide sensor. We calculate the resonant spectra for our example silicon-on-insulator hybrid plasmonic waveguide sensor, considering both changes to the sensor length and analyte index, using only propagation constant (mode) calculations. The results are verified by full-field finite element propagation calculations. This approach will be helpful in designing any analogous dielectric-plasmonic sensor, providing a first design step to identify the useful parameter range in the earliest design stage, before using detailed full propagation calculations.

## 2. Materials and Methods

Our aim is to provide a theoretical treatment for evaluating the wavelength-dependent transmission T(λ) for the prototypical configuration of Figure 2 for a given combination of $n_{a}$ and *L* based on coupled mode theory. The only parameters needed are those of the propagation constants of the participating modes of the coupled and uncoupled systems, which can be readily calculated with any reduced-dimension mode solver, accelerating computation times. In the first instance, we take all waveguides to be lossless (i.e., the permittivity of gold is taken to be the real part of its actual value [32]).

The field in an isolated dielectric waveguide—i.e., in the input section of our device (z<0 in Figure 2)—is written as the product of a mode field, which depends on the transverse coordinates and a *z*-dependent factor ψ(z)∝exp(iβz), where β is the propagation constant of the mode, ψ(z) is its amplitude, and ${\left|\psi \right|}^{2}$ is its power. The propagation in an isolated waveguide is thus described by dψ/dz=iβψ.

If two waveguides 1 and 2 are brought together (0<z<L in Figure 2) and allowed to interact linearly, then their two individual modes $\psi_{1}$ and $\psi_{2}$ couple via
(6)$$\begin{array}{c} \frac{d}{dz}\left(\begin{array}{c} \psi_{1}\\ \psi_{2} \end{array}\right)=i\left(\begin{array}{cc} \beta_{1} & \kappa \\ \kappa & \beta_{2} \end{array}\right)\left(\begin{array}{c} \psi_{1}\\ \psi_{2}, \end{array}\right). \end{array}$$
where κ is a coupling parameter, typically calculated using cumbersome overlap integrals. In general, κ is complex and the two off-diagonal elements are complex conjugates; for longitudinally invariant waveguides and in the absence of loss, the phase can be adjusted to make them both real. We note that, strictly speaking, bringing the waveguides together perturbs $\beta_{1}$ and $\beta_{2}$, and that the off-diagonal elements may differ. However, such corrections also require overlap integrals [42,43] which, as we will show, are not necessary to capture the salient modal interactions. Equation (Equation 6) can also be used to approximate dissimilar waveguides [40,44,45,46], provided that the waveguides are not too strongly coupled [42,44]. One of the goals of this work is to present the value of this simple model in the context of plasmonic waveguides, verifying its validity by direct quantitative comparisons with full calculations.

The eigenmodes of this system, also referred to as supermodes or hybrid eigenmodes, are obtained by looking for solutions of the form $\tilde{\psi_{j}}\;\exp\left(i\tilde{\beta_{j}}z\right)$, where $\tilde{\psi_{j}}$ are the eigenvectors of the matrix and $\tilde{\beta_{j}}$ are its eigenvalues (i.e., the propagation constant of each supermode). These propagation constants are given by
(7)$$\begin{array}{c} \tilde{\beta_{j}}=\bar{\beta }\pm \sqrt{\kappa^{2}+\Delta^{2}}, \end{array}$$
where $\bar{\beta }=\left(\beta_{1}+\beta_{2}\right)/2$ and $\Delta =\left(\beta_{1}-\beta_{2}\right)/2$. For identical waveguides, the mode fields of the supermodes associated with $\tilde{\psi_{j}}$ are even- and odd- superpositions of the mode fields associated with $\psi_{1}$ and $\psi_{2}$ [40].

Equation (Equation 7) immediately provides a pathway for obtaining κ accurately without overlap integrals: knowledge of the “exact” isolated- and hybrid- modes’ propagation constants, $\beta_{1,2}$ and ${\tilde{\beta }}_{1,2}$ respectively, in combination with Equation (Equation 7), yields an estimate of the coupling coefficient,
(8)$$\begin{array}{c} \kappa =\sqrt{{\tilde{\Delta }}^{2}-\Delta^{2}}, \end{array}$$
where $\tilde{\Delta }=({\tilde{\beta }}_{1}-{\tilde{\beta }}_{2}$)/2. Knowing all the parameters, and with the initial conditions $\psi_{1}\left(0\right)$ and $\psi_{2}\left(0\right)$, Equation (Equation 6) can then straightforwardly be solved to yield the transmitted power $|\psi_{i}{\left(z\right)|}^{2}$, which is a function of wavelength and device length due to mode coupling. Most importantly, this approach requires no overlap integrals at all, only knowledge of the various propagation constants—which any mode solver in reduced dimensions can provide—and access to a numerical solver of ordinary differential equations.

### 2.1. Lossless HPWG Sensor

To illustrate how the above parameters manifest in a realistic sensor, we begin by computing all eigenmodes for the device shown in Figure 2 in the absence of loss. The isolated (uncoupled) eigenmodes are calculated from equivalent dielectric waveguides without a gold film or by the gold film in the absence of silicon, as summarized at the top of Figure 3. The solid/dashed curves in Figure 3a–c show the effective index $n_{\mathrm{eff},i}=\beta_{i}/k_{0}$ of isolated/hybrid modes for analyte refractive index $n_{a}=1.3$, 1.4 and 1.5, respectively. The propagation constants are obtained by numerically solving Maxwell’s equations with suitable continuity boundary conditions for the fields at the interfaces between the layers [47]. The material dispersion for silica [33] and gold (Drude model [32]) are included, but for now we set the imaginary part of the permittivity to be zero everywhere.

We notice that the effect of increasing the analyte index $n_{a}$ is to shift the propagation constant $\beta_{2}$ to higher values. This, in turn, changes the point at which $\beta_{1}$ and $\beta_{2}$ cross: at such a point, the ${\tilde{\beta }}_{1}$ and ${\tilde{\beta }}_{2}$ anti-cross. The associated splitting of ${\tilde{\beta }}_{1,2}$ with respect to $\beta_{1,2}$ is quantified by κ via Equation (Equation 8), which in turn is plotted in Figure 3d–f. The wavelength-dependent coupling dictates the transmission spectrum, which for the lossless case has been considered extensively [22,24].

### 2.2. Lossy HPWG Sensor

We now introduce loss by numerically “switching on” the imaginary part of the gold permittivity. We now show that the loss can be accounted for by simply changing the propagation constant of $\beta_{2}$, with all other parameters remaining the same. This results in changes to the propagation constants such that $\beta_{2}=\beta_{2}^{R}+i\beta_{2}^{I}$, which we take to be the dominant perturbation, with all other parameters unchanged from the lossless case. Equation (Equation 6) then takes the form
(9)$$\begin{array}{c} \frac{d}{dz}\left(\begin{array}{c} \psi_{1}\\ \psi_{2} \end{array}\right)=i\left(\begin{array}{cc} \beta_{1} & \kappa \\ \kappa & \beta_{2}^{R}+i\beta_{2}^{I} \end{array}\right)\left(\begin{array}{c} \psi_{1}\\ \psi_{2} \end{array}\right), \end{array}$$
where κ has the same value as in the lossless case, previously obtained via Equation (Equation 8). The eigenvalues of the lossy system are still given by Equation (Equation 7), replacing $\beta_{2}\to \beta_{2}^{R}+i\beta_{2}^{I}$.

Figure 4a–c show the real part of the effective index of each mode for the lossy system, for $n_{a}=1.3$, 1.4 and 1.5, respectively. Figure 4d–f show the corresponding imaginary parts. The isolated (uncoupled) eigenmodes are again shown as dashed curves: with respect to Figure 3, we find that $ℜe(\beta_{2})$ is slightly shifted due to the perturbation introduced by loss, and $ℑm(\beta_{2})$ is non-zero, as expected. The light solid curves in Figure 4 plot ${\tilde{\beta }}_{j}$, obtained from Equation (Equation 7) using the lossy uncoupled modes $\beta_{i}$ (dashed lines in Figure 4), and the κ shown in Figure 3. The propagation constants of the two “exact” supermodes, calculated by solving the transcendental equation describing the full system, are overlayed as dark solid curves. We find that the eigenmodes obtained via this approach are in remarkably good agreement with those of the full system. Most notably, and in stark contrast to the lossless case, we observe a transition from regions where the real parts of the eigenmodes anti-cross and the imaginary parts cross ($n_{a}=1.3$ and $n_{a}=1.4$) to regions where the real parts cross and the imaginary parts anti-cross ($n_{a}=1.5$)—a feature often found in plasmonic sensors [48]. One important result in the present context is that the sensors’ eigenmode properties [25], as the analyte changes, are well predicted by the simple model presented here. In the following, we show that Equation (Equation 9) describes the properties of the full waveguide sensor and can straightforwardly be solved to rapidly estimate sensor performance over a wide range of *L*, $n_{a}$, and λ, using κ from the lossless case, and $\beta_{i}$ from the lossy case.

## 3. Results

We solve the coupled mode Equation (Equation 9) for the sensor shown in Figure 2 using the complex propagation constants $\beta_{i}$ of Figure 4 considering loss, in combination with the κ obtained from the lossless case shown in Figure 3. We take the input to be $\psi_{1}\left(0\right)=1$ and $\psi_{2}\left(0\right)=0$, corresponding to all the power being in the dielectric waveguide at input. Figure 5a shows the transmitted power $T=|\psi_{1}{\left(L\right)|}^{2}$, on a dB scale, as a function of λ and $n_{a}$ for L=10μm. In this configuration, we find a single sharp transmission resonance near λ=1.6μm and $n_{a}=1.42$, resulting from directional coupling to the plasmonic mode. To verify the validity of this model, we perform a full field finite element method (FEM) calculation (COMSOL). We use a port boundary condition at the input and output to excite and detect the fundamental TM mode of the waveguide [49]. Perfectly matched layers at every external boundary suppress any reflections in the simulation volume. We find good agreement between our FEM method and the CMT calculation, observing only a small offset in the values of λ and $n_{a}$ where the resonance is sharpest, most likely due to slight changes in the propagation constants $\beta_{1,2}$ due to the neighbouring waveguide [42]. Figure 5c,d and Figure 5e,f shows the results of the same calculation, for, respectively, L=15μm and L=20μm: with increasing *L*, a larger number of wavelength- and analyte-dependent resonances appear. These features are due to resonant interference resulting from directional coupling, which in the absence of loss occur at integer multiples *m* of the half-beat length $L_{b}=m\pi /\kappa$—longer device lengths thus allow for a wider range of *m* which satisfy this requirement. The full spectra clearly depend on the length of the device. We wish to emphasise that, because of the wavelength-dependent coupling and loss, the total transmission spectra must be calculated numerically, as we do here.

It is interesting to consider what happens for even longer *L*. Figure 5g,h show the transmittance on a dB scale, as a function of wavelength and analyte index for L=50μm using the CMT and FEM method, respectively. Note that for such a long length, even 2D finite element full-field calculations are extremely time consuming (a few minutes per individual combination of $n_{a}$ and λ on a high performance computer), due to the nanometer-scale mesh required in the gold film. Furthermore, the transmittance is <−100 dB at resonance, which is below the numerical noise of the FEM solver, and well below the signal-to-noise ratio of most spectrometers. Nevertheless, we find that the CMT and FEM methods broadly agree: many sharp resonances emerge due to a larger number of half-beat lengths supported, in the vicinity of where $ℜe(\beta_{i})$ or $ℑm({\tilde{\beta }}_{i})$ intersect. In this case, we attribute the discrepancies between CMT and FEM methods to numerical noise. The most intriguing feature, however, is that sharp resonances completely cease to exist for analyte indices $n_{a}$ above ∼1.46. In this region, the $ℜe({\tilde{n}}_{\mathrm{eff}})$ of the supermodes cross near the phase matching point—as can be seen in Figure 4c—so that the beat length is infinite, and resonances are broad and due to mode absorption only [25], associated with the blue curve in Figure 4f. Sharp resonances can only occur where mode beating is supported—i.e., where the $ℜe({\tilde{\beta }}_{i})$ anti-cross [25,48]—as can be seen in Figure 4a,b.

The boundary between regions where $ℜe({\tilde{\beta }}_{i})$ cross and anti-cross—and which thus separates regions where the detection limit of plasmonic sensors can be improved by narrow-band resonant spectra—is given by the exceptional point (EP), where the complex supermode propagation constants are degenerate, which by definition corresponds to the condition
(10)$$\tilde{\beta_{1}}-\tilde{\beta_{2}}=0.$$

According to coupled mode theory, this condition corresponds to $\kappa^{2}+\Delta^{2}=0$ [48,50], wherein the following conditions simultaneously need to be met:(11)$$\left\{\begin{array}{c} \beta_{1}-\beta_{2}^{R}=0,\\ \kappa -\beta_{2}^{I}/2=0. \end{array}\right.$$

The exceptional point is thus an important parameter for plasmonic directional couplers in general, and plasmonic sensor designs in particular, because it defines the point beyond which resonant coupling is not supported. Our formalism immediately provides a rapid way of identifying it, in terms of intuitive coupling- and loss-parameters. To illustrate this, Figure 6a shows a plot of $|\beta_{1}-\beta_{2}^{R}|/k_{0}+\left|\kappa -\beta_{2}^{I}/2\right|/k_{0}$ in the phase space considered, and which has a zero at the exceptional point as per Equation (Equation 11). A comparison with the exact supermodes is shown in Figure 6b, which plots $|\tilde{\beta_{1}}-\tilde{\beta_{2}}/|k_{0}$ as a function of $n_{a}$ and λ. An EP is found at the point where this function is zero as per Equation (Equation 10), consistently with coupled mode theory.

The above analysis shows that refractive index sensors are very sensitive to all parameters involved. At the early design stage, it is therefore essential to have rapid estimates of how transmission spectra are affected by changes in $n_{a}$, *L*, and λ. We now provide pedagogical guidelines for maximizing the performance of plasmonic sensors using simplified coupled mode theory.

## 4. Discussion

A full analysis of the above sensor—which quantifies both *S* and δλ as a function of $n_{a}$ and λ to maximize the FOM of Equation (Equation 2)—is quite laborious [29] and beyond the scope of this work. However, we can use the above formalism to provide a simple and accessible design procedure.

### 4.1. Operate at the Phase Matching Wavelength

In order to achieve resonant energy transfer between the dielectric waveguide and the gold surface, one should operate near the phase matching (PM) wavelength $\lambda_{PM}$ where $\beta_{1}\left(\lambda_{\mathrm{PM}}\right)=\beta_{2}^{R}\left(\lambda_{\mathrm{PM}}\right)$, identified, for example, as the wavelength where the dashed lines in Figure 4a–c intersect for $n_{a}=1.3,1.4,1.5$, respectively. Figure 7a (right axis, green circles) shows a detailed plot of the calculated $\lambda_{\mathrm{PM}}$ as a function of analyte index $n_{a}$ in the present configuration, which provides the first estimate of where resonances are expected for different choices of $n_{a}$.

### 4.2. Calculate the Nominal Sensitivity

The phase matching wavelengths also provide a first estimate of the sensitivity, $S_{\mathrm{PM}}=d\lambda_{\mathrm{PM}}/dn_{a}$. This can be evaluated early on, before proceeding with full calculations of the resonant transmission spectra to obtain δλ and the FOM. In the present configuration, $S_{\mathrm{PM}}=-2340\;\mathrm{nm}/\mathrm{RIU}$, obtained from a linear fit to the data in Figure 7a. Note, however, that for short device lengths this value can differ significantly, as we show below.

### 4.3. Operate above the Exceptional Point

Equation (Equation 11) indicates that a sensor should also satisfy $\kappa -\beta_{2}^{I}/2>0$ at $\lambda_{\mathrm{PM}}$. This yields the condition for the real parts of the supermodes to split at the phase matching point, so that the device can harness the sharp resonances induced by modal beating, rather than the broad resonances induced by metallic losses. In the present configuration, this occurs for $n_{a}<1.46$.

### 4.4. Identify the Nominal Device Length

Calculations of the eigenmodes using mode solvers can be used to provide a first estimate of the shortest device length over which resonant energy transfer occurs due to modal beating, which corresponds to the half beat length,
(12)$$L_{b}=\frac{\pi }{ℜe(\tilde{\beta_{1}}-\tilde{\beta_{2}})},\;$$
also calculated at the phase matching wavelength $\lambda_{\mathrm{PM}}$. Furthermore, because the modes are lossy, device length should be kept short to avoid the resonances to fall below the instrument noise—ideally not much longer than each supermodes’ absorption lengths,
(13)$$L_{a}^{i}=\frac{1}{2ℑm(\tilde{\beta_{i}})},\;$$
which provide an estimate of the length scale over which the power in each mode decays by a factor of 1/e. Figure 7a (left axis) shows the calculated $L_{b}$ as a function of analyte index $n_{a}$, which is in the range of 2–10 μm. The associated average $L_{a}$ of the two supermodes is shown in Figure 7b. Note that the beat length is here comparable to (or longer than) the absorption length, making short devices necessary for practical applications.

### 4.5. Calculate the FOM

Following the above calculations, we proceed with calculating the FOM of the full device. For the present example, we consider L=10μm. For this device length, we inspect the onset of a narrow linewidth δλ in the transmission spectrum as a result of resonant coupling, leading to the highest accessible FOM. We highlight $L_{b}=10\;\mu m$ as a black dashed line in Figure 7a: the resonance will occur near a wavelength 1600 nm and analyte index $n_{a}=1.44$.

#### 4.5.1. “Conventional” Mode Approach

Before proceeding with the FOM obtained from CMT, it is worth discussing the results obtained using common mode-based approaches in regions where avoided crossings (in terms of $ℜe({\tilde{\beta }}_{i})$) occur. This “conventional” mode approach is used, e.g., in Refs. [51,52,53,54,55], and appears frequently throughout the plasmonic sensing literature: it attributes the absorption spectrum to the supermode with the lowest loss at a given wavelength. In practical terms, the transmitted power (in dB) is taken to be
(14)$$T=10\log_{10}\exp\left\{-2L\times \min\left[ℑm\left({\tilde{\beta }}_{1}\right),ℑm\left({\tilde{\beta }}_{2}\right)\right]\right\},\;$$
i.e., the loss is computed from the minimum of the two hybrid mode loss curves of Figure 4d–f, and both $\lambda_{R}$ and δλ follow immediately from $ℑm({\tilde{\beta }}_{i})$. The resulting transmission spectra for the plasmonic sensor presented here, using Equation (Equation 14), are shown in Figure 8a for different analyte indices as labelled. The resonant wave length $\lambda_{R}$ is readily identified and plotted as a function of analyte index in Figure 8b (green points, left axis). The corresponding sensitivity (orange line, right axis) is obtained from the derivative of a second-order polynomial fit (green line). In addition, the associated characteristic width δλ, taken as the FWHM with respect to the minimum transmission (i.e., the spectral width at twice the minimum transmission, 3 dB above the transmission minimum), is shown as an orange line in Figure 8c (right axis). This yields the FOM via Equation (Equation 2), shown as a green line in Figure 8c. Note in particular that δλ decreases linearly with $n_{a}$, as the exceptional point is approached. These results are all independent of *L*, in contrast with our earlier analysis (cfr. Figure 5).

#### 4.5.2. Coupled Mode Theory Approach

We now compute the FOM from the CMT method for L=10μm, quantitatively analyzing the spectra which produce the colour map in Figure 5a. Figure 8a plots the spectra at the analyte index as labelled. Note the important differences with respect to the conventional case in Figure 8. First, a plot of $\lambda_{R}$ vs. $n_{a}$, shown as green circles in Figure 8e (left axis), indicates a sensitivity which is several times smaller (orange line, right axis). Second, the minimum δλ does not increase towards the exceptional point, but has a local minimum near $n_{a}=1.42$, due to resonant coupling, as shown in the orange curve of Figure 8f (right axis). As a result of this narrow linewidth, however, the FOM reaches values of up to ∼100, as illustrated in the green curve of Figure 8f (left axis). This is broadly consistent with the same analysis using finite element calculations, shown in Figure 8g–i—the only difference being a small shift in the analyte index where the high FOM occurs. Therefore, we expect our CMT approach to be a valuable first step in device design, providing an estimate of the plasmonic sensor performance using low computational resources and simple modal parameters, which should ideally be followed by a full detailed calculations in the parameter subspace of interest, e.g., via finite element [49], or eigenmode [25,29] calculations.

### 4.6. Towards Optical Fibre Plasmonic Sensors

The CMT formalism is agnostic to the waveguide geometry used, and can in principle be applied to fibre plasmonic sensors [20]. In fibre-based structures, the dielectric mode is typically found within a micrometer-scale silica core [16,18], and the plasmonic mode is typically guided by a metallic film in its vicinity [16]. Compared to the structure presented here, therefore, we believe that fibre-based structures would present three important additional features which should be accounted for in future investigations.
The present dielectric waveguide is formed by a high-index, sub-wavelength silicon core and a silica cladding: its higher propagation constant provides access to the short-range SPP, which is supported at all wavelengths shown and does not cut off. In contrast, fibre plasmonic sensors typically use a wavelength-scale lower-index silica (SiO_2_) core, wherein the effective index of the dielectric mode is close to the refractive index of silica ($n_{\mathrm{eff}}\approx n_{\mathrm{SiO}2}=1.45$). This mode typically phase-matches to the weakly confined long-range surface plasmon (LR-SPP) [56] for an analyte refractive index close to $n_{a}=1.45$, and typically cuts off close to regions where the supermodes anti-cross [48]. High-order plasmonic modes in metallic nanowires also cut off across the visible and infrared spectrum [57,58]. The present formalism can only only be applied in regions of the parameter space where the uncoupled bound states are supported, i.e., below modal cutoff.The present plasmonic sensor is a two-mode system, because each uncoupled waveguide is single mode. Fibre plasmonic sensors, on the other hand, typically have core sizes of several wavelengths in diameter, and can be highly multi-mode. In multi-mode dielectric fibres, the dimensions of the matrix in Equation (Equation 6) must therefore be increased to account for the additional modes and coupling coefficients [59].Finally, we wish to point out that, in order to achieve sharp resonances and high FOMs in multi-mode sensors, a single-mode waveguide/fibre at input- and output- is required, which filters out higher-order modes, because these have the effect of washing out sharp resonant dips and lowering the FOM [25,48].

## 5. Conclusions

In conclusion, we have developed a simplified lossy coupled mode theory which obtains coupling coefficients from lossless waveguides, and subsequently introduces loss as a perturbation. This formalism predicts where the real parts of the coupled eigenmodes cross and anti-cross, and how this quantitatively impacts plasmonic sensors’ transmission spectra, as validated by full-field calculations. Our approach lends itself to a wide class of sensor structures [16,38,60,61,62,63,64,65,66,67,68,69,70,71,72,73] and we expect it to be used as a valuable first step in rapidly estimating the energy transfer properties of many hybrid plasmonic waveguide systems, using limited computational resources and easily obtainable modal parameters. Note that the present CMT approach is only valid in regions where the coupling between waveguides is not too strong. Preliminary investigations suggest that deviations from the full field calculations start to become significant when $\kappa /|\tilde{\Delta }|≳0.1$. A detailed analysis of the limits of this method will be the subject of future work.

## Figures and Tables

**Figure 1 sensors-22-09994-f001:**
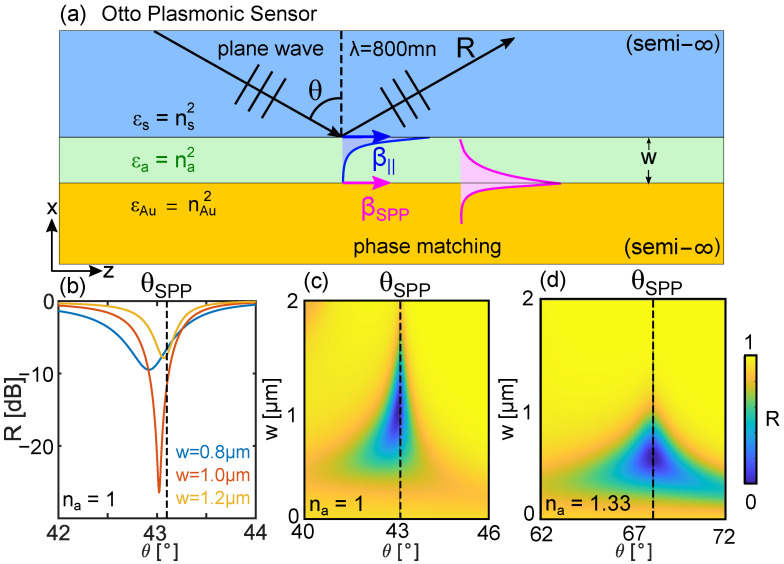
Concept schematic of the challenge of calculating resonances in plasmonic sensors. (**a**) The simple Otto configuration relies on monitoring the reflectivity *R* of plane waves propagating in semi-infinite media as a function of angle θ. At the angle $\theta_{\mathrm{SPP}}$ a SPP is excited. (**b**) θ-dependent reflectance spectrum for $n_{a}=1$, λ=800nm, and *w* as labelled. Also shown is the full colourmap of the reflectance as a function of θ and *w* for (**c**) $n_{a}=1$ and (**d**) $n_{a}=1.33$. Note that the spectral maps are subtly dependent on both $n_{a}$ and *w*.

**Figure 2 sensors-22-09994-f002:**
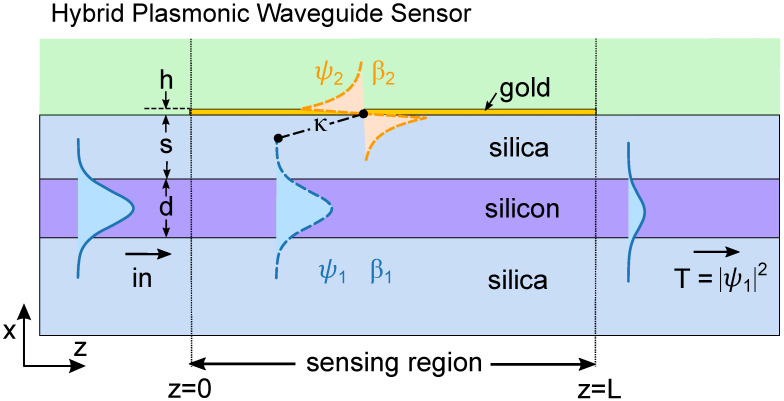
Schematic of the HPWG sensor and the coupled mode theory picture. The modes in the dielectric and plasmonic regions, $\psi_{1}$ and $\psi_{2}$ respectively, couple linearly as described by Equation (Equation 6). The power in the dielectric at output is given by $T=|\psi_{1}{|}^{2}$. The periodic exchange of power between waveguides can lead to a resonant spectrum that in general depends on both the length of the device *L* and the analyte index $n_{a}$ [25].

**Figure 3 sensors-22-09994-f003:**
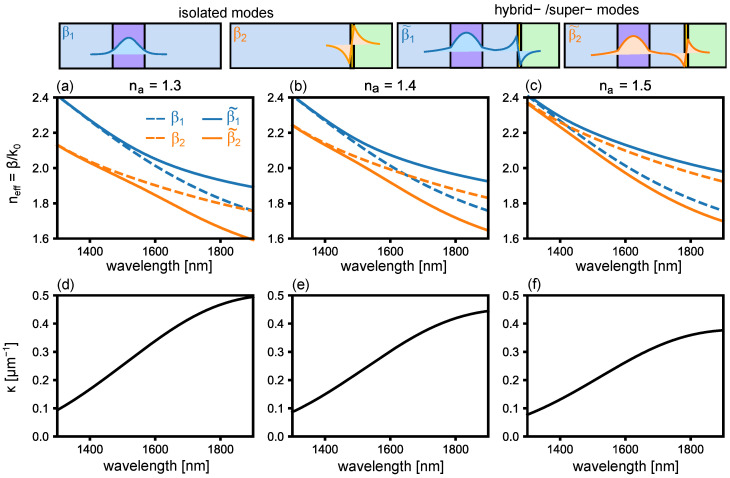
Effective index $n_{\mathrm{eff}}=\beta /k_{0}$ as a function of wavelength for the geometry shown in Figure 2 when (**a**) $n_{a}=1.3$, (**b**) $n_{a}=1.4$, (**c**) $n_{a}=1.5$ in the lossless case. The dashed line shows the isolated plasmonic- and dielectric- modes, respectively. The solid lines show the hybrid eigenmodes. (**d**–**f**) show the associated calculated coupling coefficients, following the simple expression in Equation (Equation 8) (black line). Top row shows a schematic of the magnetic field for the plotted isolated- or hybrid-/super-modes.

**Figure 4 sensors-22-09994-f004:**
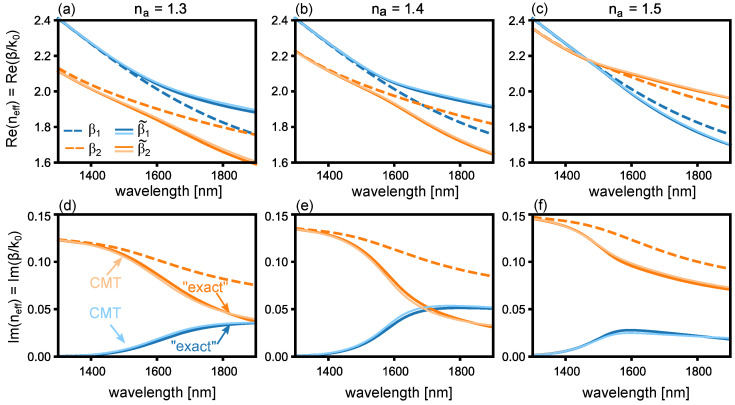
Real part of the effective index $ℜe\left(n_{\mathrm{eff}}\right)=ℜe\left(\beta /k_{0}\right)$ as a function of wavelength for the geometry shown in Figure 2 when (**a**) $n_{a}=1.3$, (**b**) $n_{a}=1.4$, (**c**) $n_{a}=1.5$, using the lossy Drude model for the gold permittivity. The dashed line shows the isolated plasmonic- and dielectric- modes, respectively. The solid lines show the hybrid eigenmodes according to the “exact” solution (dark) and obtained from CMT via the eigenvalues of Equation (Equation 9) (light). (**d**–**f**) show the associated $ℑm(n_{\mathrm{eff}})$.

**Figure 5 sensors-22-09994-f005:**
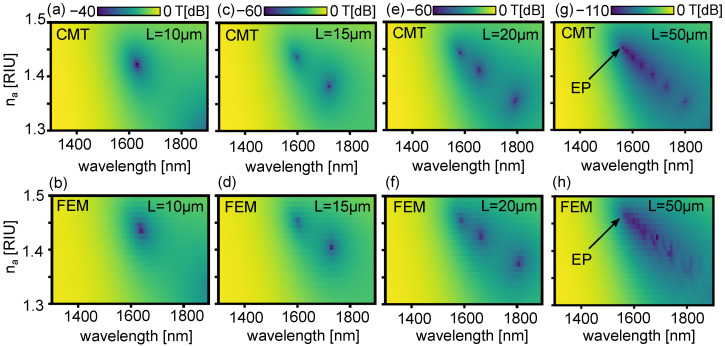
Transmitted power by the plasmonic sensor as a function of λ and $n_{a}$ for L=10μm using (**a**) CMT, and (**b**) FEM. (**c**,**d**): same as (**a**,**b**) for L=15μm. (**e**,**f**): same as (**a**,**b**) for L=20μm. (**g**,**h**): same as (**a**,**b**) for L=50μm. EP: exceptional point.

**Figure 6 sensors-22-09994-f006:**
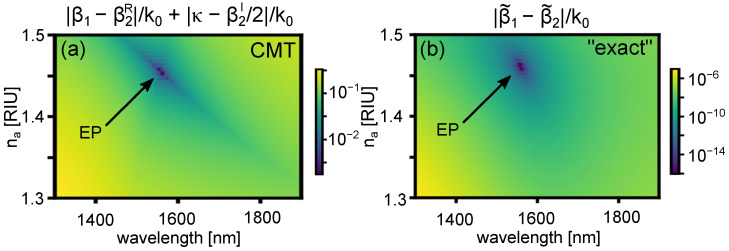
Calculated colour maps of (**a**) $|\beta_{1}-\beta_{2}^{R}|/k_{0}+\left|\kappa -\beta_{2}^{I}/2\right|/k_{0}$ using CMT and (**b**) $|\tilde{\beta_{1}}-\tilde{\beta_{2}}|/k_{0}$ using the exact supermodes. The global minima in the phase space show the location of the exceptional point using our CMT model and the exact solution, as per Equations (Equation 10) and (Equation 11).

**Figure 7 sensors-22-09994-f007:**
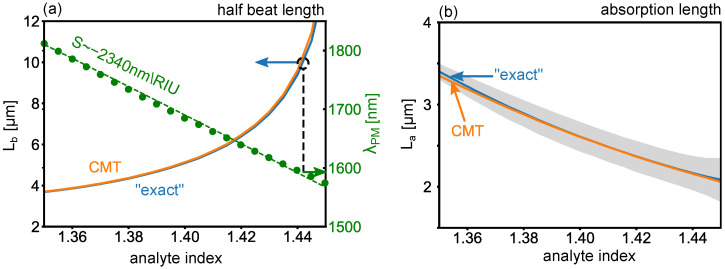
(**a**) Green (right axis): phase matching wavelength $\lambda_{\mathrm{PM}}$ where $\beta_{1}=\beta_{2}^{R}$, and associated half beat length $L_{b}$ according to the supermodes obtained with CMT (orange) and “exact” calculations (blue). (**b**) Associated absorption length $L_{a}$. Solid lines indicate the average $L_{a}=\left(L_{a}^{1}+L_{a}^{2}\right)/2$; shaded regions encompass the $L_{a}^{1}$ and $L_{a}^{2}$ boundaries.

**Figure 8 sensors-22-09994-f008:**
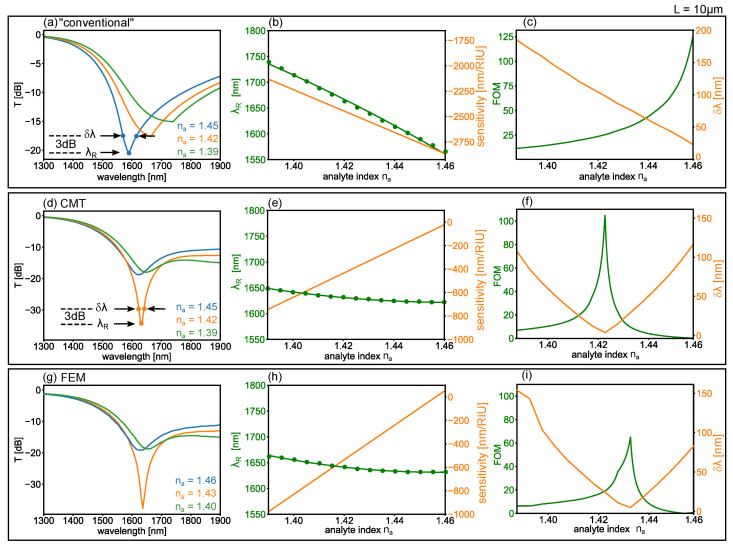
(**a**) Transmission spectrum using the “conventional” approach of Equation (Equation 14), as a function of wavelength, for the three analyte indices as labelled, using L=10μm. Also shown are the resonant wavelength $\lambda_{R}$, corresponding to the spectral minimum and the δλ, corresponding to the FWHM. (**b**) Associated $\lambda_{R}$ vs. $n_{a}$ (green circles, left axis), second order polynomial fit (green line), and resulting sensitivity *S* (orange line, right axis.) Also shown in (**c**) are the δλ vs. $n_{a}$ (orange curve, left axis) and the total FOM = S/δλ. (**d**–**f**): same as (**a**–**c**), obtained from the CMT approach, using a subset of the data shown in Figure 5a as labelled. (**g**–**i**): same as (**d**–**f**), obtained from FEM calculations, using a subset of the data shown in Figure 5b as labelled.

## Data Availability

The data and code that support the findings of this study are available from https://github.com/tuniz/sensors (accessed on 14 December 2022).

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
