# Peer review of "Plasmonic Sensors beyond the Phase Matching Condition: A Simplified Approach"

_sensors, 2022, doi:10.3390/s22249994_

Round 1
Reviewer 1 Report
Plasmonic sensors are attractive for nowdays. This manuscript gives a useful framework for dielectric-plasmonic sensor design by t a simplified coupled mode theory. I hope for its publication and it will be helpful for the researchers in this field.
Author Response
We thank the reviewer for the positive comments. We welcome specific recommendations for improvement.
Reviewer 2 Report
This paper addresses the surface plasmon polariton resonances problems in waveguides. The paper is very well written and will be a valuable tool for researchers working in the field. It can be published in the present form.
Author Response
We thank the Reviewer for the positive comments.
Reviewer 3 Report
In the present work, the authors propose a method for the design of waveguide-based surface plasmon resonance sensors including a single-mode, SMF, and multimode fiber, MMF, optic sensor (although it would be interesting if they had included an optical fiber sensor with a different core (MMF-SMF-MMF).
The analysis developed in the introduction of the Otto configuration seems irrelevant since it is a configuration of little practical use due to the fact that the air gap is very small. Assuming that the separation is 1000nm, the surfaces of the prism and the metal thin film are required to be ultra-flat. On the other hand, it would have been interesting if this analysis had been developed for the Kretschmann configuration, one of the most popular configurations due to its practicality and ease.
I consider that the manuscript does not need any modifications and is sufficient for its publication.
Author Response
We thank the Reviewer for the positive comments, and that no modifications are required.
We would like to take the opportunity to comment that the well-known Otto configuration is used an example to show that coupling to surface plasmons is dramatically dependent on small configuration changes, even in such a simple case. This feature is best exemplified by the Otto configuration, where the plasmonic resonance subtly depends on the separation between the gold and the dielectric.
We would also like to note that, since this is the first time (to our knowledge) that this simplified coupled mode formalism is used to treat plasmonic waveguide sensors, we have started with a two-mode system. Future studies should generalize this to multi-mode systems, as pointed out at the and of Page 12:
"The present plasmonic sensor is a two-mode system, because each uncoupled waveguide is single mode. Fiber plasmonic sensors, on the other hand, typically have core sizes of several wavelengths in diameter, and can be highly multi-mode. In multi-mode dielectric fibers, the dimensions of the matrix in Eq. (6) must therefore be increased to account for the additional modes and coupling coefficients [59]."